# GRIFFIN: GENERATIVE REFERENCE AND LAYOUT GUIDED IMAGE COMPOSITION

griffin-anonymous.github.io

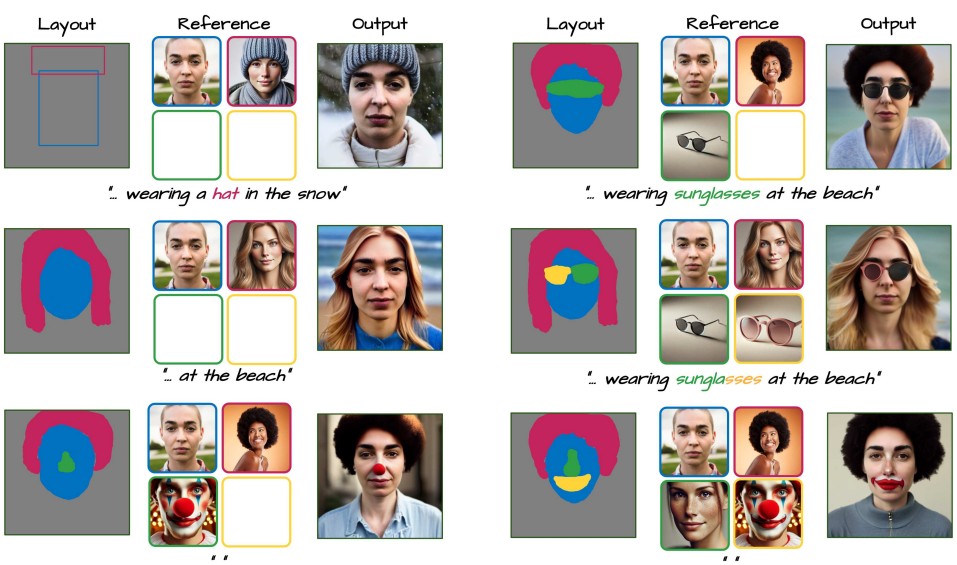

**Figure 1:** With Griffin, we can generate an image by defining both the content to be incorporated and its placement within the final composition. By conditioning on different images and specifying layouts using either bounding boxes or pixel masks, our method enables a wide range of compositional variations. The base prompt is "A portrait of a woman ...".

## ABSTRACT

Text-to-image models have reached a level of realism that enables highly convincing image generation. However, text-based control can be a limiting factor when more explicit guidance is needed. Defining both the content and its precise placement within an image is crucial for achieving finer control. In this work, we address the challenge of multi-image layout control, where the desired content is specified through images rather than text, and the model is guided on where to place each element. Our approach is training-free, requires a single image per reference, and provides explicit and simple control for object and part-level composition. We demonstrate its effectiveness across various image composition tasks.

## 1 INTRODUCTION

Diffusion-based text-to-image models excel at generating diverse and intricate visuals, ranging from realistic scenes to abstract compositions. While they offer impressive versatility, achieving precise control over the final output (both in terms of which visual content to include, and where it will be placed) is essential for aligning the generated image with the user's intent. To enhance this control, a composition technique that seamlessly integrates elements from different images and arranges them cohesively, guided by specific hints or instructions, is highly valuable.

> *"Griffin: a mythical creature with the head and wings of an eagle and the body of a lion, and with the eagle's legs taking the place of the forelegs."*  – New Oxford American Dictionary

Inspired by the legendary creature, we introduce *Griffin*: a method that enables the precise combination of parts or subjects from different images, placing them in locations specified by the user. This task is challenging as it requires seamless blending of elements to form a realistic composition, while ensuring that the subjects are reproduced faithfully; see Figure 1. There are two key aspects of image generation over which we want to exert explicit control:

- *Identity preservation:* text can only provide a loose description of the image content, we would like to be able to cue the generator using example images, rather than text, and we would like the identity/style of the content within these images to be preserved as much as possible in the generated images.

- *Layout specification:* precisely defining the placement of content within an image with text is challenging, and artists typically use visual mock-ups rather than textual descriptions to communicate a scene's layout effectively.

The importance of personalized images and layout control has been recognized in previous work (Gal et al., 2022; Ruiz et al., 2023; Kumari et al., 2023; Dahary et al., 2024; Li et al., 2023b; Jang et al., 2024; Liu et al., 2023; Tarrés et al., 2025). However, these methods are unable to perform training-free part-level composition effectively, as the identities of multiple parts tend to leak together. To capture the identity of a concept, they require multiple images per subject and lengthy training to optimize and learn a token for each concept.

Our approach transfers appearance from relevant pixels in the source images using an attention-sharing mechanism, previously applied in image editing (Cao et al., 2023). Attention sharing alone does not natively support layout control. Relying on text prompts also does not guarantee accurate placement or adherence to the specified layout (Figure 2-a). Since the initial Gaussian noise in text-to-image models contains spatial information, using inversion to start generation from the inverted noise can produce realistic results. However, it heavily constrains the structure to the input image rather than allowing flexibility based on the specified layout or text (Figure 2-b).

To align with the provided layout, we use an encoder-based personalization method such as IP-Adapter (Ye et al., 2023) to anchor each layout component to its corresponding source image. However, IP-Adapter independently does not fully preserve the identity and fine details of the subjects (Figure 2-c). To address this, we first use IP-Adapter to establish a correct structure in the early denoising steps, ensuring a strong foundation for further refinement. Afterward, we introduce a layout-controlled attention-sharing mechanism, where each image patch derives its appearance either from its corresponding reference image or the text prompt, depending on whether it belongs to a layout component or the background. This way, the appearance of the source images is preserved. In addition, part-level composition is enabled without appearance leakage by ensuring that each patch attends only to its corresponding source image and relevant regions within the target image (Figure 2-d and Figure 3-c).

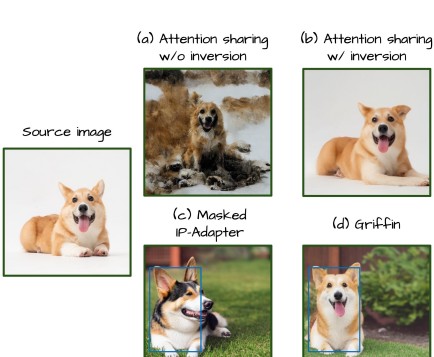

**Figure 2: Control in image generation** – Naïve attention-sharing lacks explicit layout control. (a) and (b) are generated using the text: "A dog sitting in the yard. The dog is on the left side of the image." but fail to reliably position the subject. In (c), masked IP-Adapter is used, but it struggles with identity preservation. (d) shows our method, which successfully maintains the subject's identity and adheres to the layout and text prompt.

Our method allows for precise control over the appearance and layout of the image. Griffin needs *only one* reference image per subject and supports both object-level and part-level composition. We demonstrate the effectiveness of our method across a range of image composition tasks, showing both quantitatively and qualitatively that it outperforms the state-of-the-art.

## 2 RELATED WORKS

Recent advances in large-scale diffusion models (Dhariwal & Nichol, 2021; Ho et al., 2020) have greatly enhanced the variety and quality of visual content. Leveraging free-form text (Balaji et al., 2023; Ramesh et al., 2022; Rombach et al., 2022; Saharia et al., 2022), these models can generate multiple concepts within a single image. Despite their high expressiveness, they do not inherently support user-defined concepts or spatial guidance, motivating further research on spatially guided image generation and personalization for diffusion models.

**Spatial guided image generation.** While text prompts can effectively describe high-level semantics, they often lack sufficient control over spatial arrangements in image generation. To address this limitation, additional guidance such as segmentation maps (Zeng et al., 2022), depth maps (Eldesokey & Wonka, 2024; Jo & Choo, 2024), sketches (Zhang et al., 2024; Mikaeili et al., 2023), and bounding boxes (Zheng et al., 2024; Yang et al., 2022; Zhou et al., 2024; Tarrés et al., 2025) has emerged. These spatial cues help ensure objects appear with the correct placement and size. ControlNet (Zhang et al., 2023b) incorporate structural signals (e.g. edges, poses) for even finer spatial fidelity. Since compositional image generation can be subjective, we incorporate layout components (e.g., masks, bounding boxes) to ensure greater control over the placement and structure of the generated content.

**Personalization in text-to-image models.** Personalized text-to-image generation focuses on adapting a pre-trained generative model so that it can create novel images of a specific concept, subject, or style, supplied by a small number of reference images. Finetuning-based methods (Gal et al., 2022; Ruiz et al., 2023; Kumari et al., 2023; Alaluf et al., 2023b; Safaee et al., 2023) update the network parameters and texual embeddings to capture a personalized concept while balancing subject fidelity and prompt-driven variability. Alternatively, training-free personalization methods inject references directly into the generation process. IP-Adapter (Ye et al., 2023), for example, extracts image features using a projection layer and applies them through cross-attention to guide generation according to the reference image. However, our observations indicate that while IP-Adapter is effective at capturing global structural and appearance attributes, it struggles with fine-grained details. Most recently, Multiwine (Tarrés et al., 2025) introduced a multi-concept localized generation, injecting reference image features through cross-attention and encoding layout by concatenating masks with the noisy latent. For this, they carefully curate a dataset and finetune the stable diffusion inpainting model Rombach et al. (2022) to accept image and conditions. However, as we show in our supplementary, their method does not preserve the identity of reference objects, similar to encoder-based approaches such as IP-Adapter.

**Multi-concept personalization.** An additional line of research explores decomposing and recomposing multiple personalized concepts within a single generated image. Several approaches adapt embeddings or model weights to incorporate new concepts (Kong et al., 2024; Patashnik et al., 2025a; Shi et al., 2023; Po et al., 2024; Yang et al., 2024; Kumari et al., 2023; Jang et al., 2024). (Avrahami et al., 2023; Garibi et al., 2025) introduce a notion of extracting separate tokens for each object in a scene, enabling new re-compositions, whereas methods such as (Gu et al., 2023; Liu et al., 2023) adopt more spatially guided generation strategies for combining multiple concepts. (Parmar et al., 2025) trains a two-level coarse and fine encoder for object-level scene composition. However, these approaches generally require additional training or fine-tuning steps and cannot achieve the fine-grained, part-level composition. In contrast, our method requires no additional training.

**Attention-based identity preservation.** Maintaining the identity of a subject while altering layouts or scenes can be addressed with attention-sharing. (Cao et al., 2023; Mou et al., 2023) propose querying correlated local contents and textures from source images for editing, ensuring consistency in appearance. Moreover, (Alaluf et al., 2023a) uses this attention-sharing mechanism for appearance transfer, and (Hertz et al., 2024) applies it for style-transfer. Similarly, sharing self-attention keys and values of the first frame across subsequent frames has been used to improve temporal consistency in video generation (Wu et al., 2023; Ceylan et al., 2023; Khachatryan et al., 2023), while also facilitating consistent video editing (Geyer et al., 2023; Qi et al., 2023). While these methods simply concatenate the keys and values across different images or frames, (Deng et al., 2023) propose a weighted attention mixing method that can focus more on the source image while generating newly added regions using the target image. Generative photomontage (Liu et al., 2025)

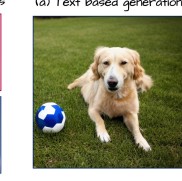 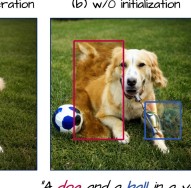 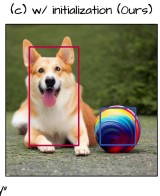

Source images    (a) Text based generation    (b) w/o initialization    (c) w/ initialization (Ours)

*"A dog and a ball in a yard"*

**Figure 3:** Without proper initialization, attention sharing generates an image as if attention sharing were absent, leading to artifacts (a), (b). Using masked IP-Adapter for initialization allows attention sharing to effectively transfer appearance from the sources to each subject in the target (c).

proposes mixing queries, keys, and values of images which are structurally aligned for appearance composition. Most recently, NestedAttention (Patashnik et al., 2025b) trains an encoder to learn a per-patch value token in the cross-attention modules for fine-grained identity preservation. Alternatively, our method uses IP-Adapter Ye et al. (2023) to set the global composition and appearance of the scene, and performs attention-sharing in the later steps of the diffusion process to improve identity preservation.

## 3 PRELIMINARIES

We first provide an overview of text-to-image model architectures (Rombach et al., 2022). At timestep $t$, a noisy image $x_t$ is passed through the diffusion model to denoise it, producing $x_{t-1}$. The denoising architecture consists of multiple layers. At each network layer $l$, a self-attention module, followed by a cross-attention module, conditions the generation on the input text-prompt. The input to the attention layer is the intermediate feature map $h_l$. This feature map is linearly projected into queries ($Q$). The keys ($K$) and values ($V$) are obtained by projecting a feature sequence $f_l$, which in self-attention is equal to $h_l$, and in cross-attention, it is the text token embeddings.

$$Q = W_l^Q h_l, \quad K = W_l^K f_l, \quad V = W_l^V f_l. \tag{1}$$

The attention module output would be:

$$f_A = \text{Attention}(Q, K, V) = A \cdot V, \tag{2}$$

where $A$ is the attention matrix computed as:

$$A = \text{Softmax}\left(Q^T K / \sqrt{d}\right), \tag{3}$$

and $d$ is the feature dimension of $Q$ and $K$. To establish notation, we now quickly review the identity preservation and image-conditioned generation methods.

**Attention sharing.**

Recent works (Cao et al., 2023; Hertz et al., 2024; Alaluf et al., 2023a) show that allowing image features to attend to source image keys and values during denoising aids identity preservation. MasaCtrl (Cao et al., 2023), in particular, proves this method effective for maintaining image appearance in text-guided editing. Their approach first *inverts* the denoising process to obtain a noise image (Song et al., 2022), caching the self-attention keys ($K_S$) and values ($V_S$) from this step. Then, using the inverted noise and an editing prompt, they generate the target image by replacing self-attention keys and values with $K_S$ and $V_S$. The self-attention output becomes:

$$A_S \cdot V_S, \quad A_S = \text{Softmax}\left(Q^T K_S / \sqrt{d}\right) \tag{4}$$

**IP-Adapter.** To enhance control beyond text prompts, several works have extended text-to-image models to be conditioned on input images (Ye et al., 2023; Li et al., 2023a; Gal et al., 2024). In particular, IP-Adapter (Ye et al., 2023) introduces additional cross-attention modules that condition generation on image tokens. These tokens are obtained by encoding the image with a pre-trained image encoder (Radford et al., 2021) to extract a global image feature, which is then processed by a small adapter network. The resulting image tokens are incorporated into the generation process by modifying the cross-attention mechanism. Specifically, the outputs of the text and image cross-attention modules are combined in the attention matrix: $f_{CA} = A_{\text{text}} \cdot V_{\text{text}} + s A_{\text{image}} \cdot V_{\text{image}}$, where $A_{\text{image}}$ and $V_{\text{image}}$ are attention maps and values of the added image cross attention obtained from the image embeddings and $s$ is a scalar controlling the influence of the input image on the generation.

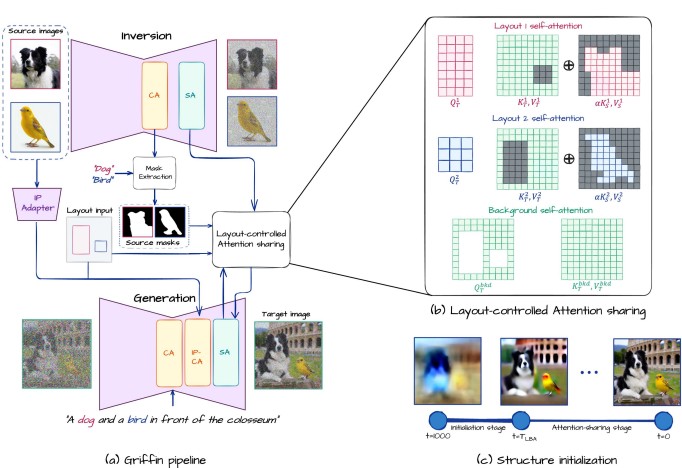

**Figure 4: Pipeline** – (a) We use IP-Adapter to initialize the structure of the target image based on the layouts. We then apply our layout-controlled attention sharing. (b) Our attention-sharing mechanism allows our generator to only attend to sub-portions of the input images, avoiding identity leakage. (c) We apply masked IP-Adapter with a high scale at the initialization stage to rapidly align the image with the input layout. At timestep $T_{LBA}$, attention-sharing begins, and the IP-Adapter scale is reduced. The displayed images are the denoised predictions at timesteps 1,000, $T_{LBA}$ and 0.

## 4 METHOD

Given a set of layouts $M=\{M_1, M_2, \ldots, M_N\}$ and corresponding source images $I_S=\{I_S^1, I_S^2, \ldots, I_S^N\}$, our goal is to generate a target image $I_T$ that respects the spatial arrangement of $M$, while, at the same time, preserving the appearance of the source images. Our layout can be specified via image masks or via bounding boxes. As layouts specified via bounding boxes are just converted to masks, in what follows with $M$ we always refer to image masks.

**Outline.** Naïvely applying the attention-sharing mechanism leads to unintended appearance copying and artifacts, as it can be observed in Figure 3-b. As we aim to generate an entirely new layout, proper initialization is crucial to ensure that features in the target image attend to the "correct" regions of the source images. We address this shortcoming by dividing the generation process into two stages. In the first stage (Section 4.1), we use an encoder-based personalization method to initialize the overall structure of the generated image. In the second stage (Section 4.2), we apply a layout-controlled attention-sharing, allowing pixels within each layout component to attend to their corresponding source image. Further, as the user-specified target layouts only coarsely represent the structure of the target image, we update the layout masks as the layout initialization is generated, leading to a significant boost in generated image quality (Section 4.3). An overview of our model architecture can be found in Figure 4.

### 4.1 STRUCTURE INITIALIZATION

To generate image $I_T$, we align the features of each layout component $M_n$ with its corresponding source image $I_S^i$, ensuring effective appearance transfer through attention-sharing. To achieve this, we use a masked IP-Adapter cross-attention mechanism, hence conditioning each region $M_n$ separately. Specifically, for each layout component $M_n$, the cross-attention output is given by:

$$f_{\text{CA}} = A_{\text{text}}V_{\text{text}} + s\sum_{n=1}^{N} M_n \odot A_{I_S^n}V_{I_S^n}, \qquad \text{where } A_{I_S^n} = \text{Softmax}\left(Q^T K_{I_S^n}/\sqrt{d}\right),$$

and $\odot$ is an element-wise product, $K_{I_S^n}$ and $V_{I_S^n}$ are the keys and values derived from the image tokens of $I_S^n$ via the IP-Adapter image encoder and adapter network. During denoising, we initially set a high scale $s$ to rapidly align the features of $I_T$ with the source images. As the process transitions to the attention-sharing stage at timestep $T_{LBA}$, $s$ is gradually reduced stepwise (Figure 4-c).

### 4.2 LAYOUT-CONTROLLED ATTENTION-SHARING

To obtain the noise representation of each source image, we first apply DDIM inversion (Song et al., 2022) to the source images. During this process, we cache the keys and values from the self-attention modules, which will be used for attention-sharing.

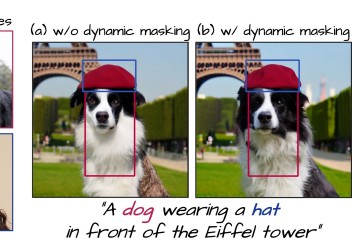

**Figure 5: Dynamic layout update** – We extract DIFT (Tang et al., 2023) and DINO (Caron et al., 2021) features from the source and target images, then compute pixel correspondences following (Zhang et al., 2023a). We discard pixels without correspondence and group the remaining pixels by their corresponding source image. Farthest sampling is used to obtain subject-specific group points, which are then fed into SAM (Kirillov et al., 2023) to generate updated masks.

**Source masks.** To extract source masks that isolate the desired subject in each source image, we leverage the cross-attention maps obtained in the inversion of the source images. Specifically, we use the cross-attention of the text token corresponding to the desired region, as proposed in prompt-to-prompt (Hertz et al., 2022). This results in a set of source masks, denoted as $\{M_S^1, M_S^2, \ldots, M_S^N\}$, where each $M_S^n$ selects the relevant region in the source image $I_S^n$.

**Attention sharing – Figure 4-b.** Each self-attention module first partitions the target query map $Q_T$ into $\{Q_T^1, \ldots, Q_T^N, Q_T^{\text{bkd}}\}$, where $Q_T^{\text{bkd}}$ corresponds to the background queries. For the $n$-th layout component, self-attention is then computed as:

$$f_{\text{SA}}^n = \text{Attention}(Q_T^n, \hat{K}_n, \hat{V}_n), \tag{5}$$

$$\hat{K}_n = \alpha \cdot (M_S^n \otimes K_S^n) \oplus K_T^n, \tag{6}$$

$$\hat{V}_n = (M_S^n \otimes V_S^n) \oplus V_T^n, \tag{7}$$

where $\otimes$ extracts only the masked features, and $\oplus$ represents concatenation. The terms $K_T^n$ and $V_T^n$ are the keys and values of the target image, restricted to the pixels of layout component $n$ and the background pixels. The parameter $\alpha$ controls the extent of appearance transfer from the source images to the target image. Similarly, for the background self-attention, we have:

$$f_{\text{SA}}^{\text{bkd}} = \text{Attention}(Q_T^{\text{bkd}}, K_T, V_T). \tag{8}$$

Intuitively, in our layout self-attention mechanism, each target pixel attends to other pixels within the same layout component, the corresponding regions in the source image, as well as the background pixels. Meanwhile, background pixels attend to all pixels in the target image, as the background generation is mostly driven by text conditioning.

### 4.3 DYNAMIC LAYOUT UPDATE

Since we do not require the user to provide precise masks, the generated content can span beyond these coarse layouts. To avoid the foreground from leaking into background areas (Figure 6) and to enforce identity preservation outside the coarse layouts, we

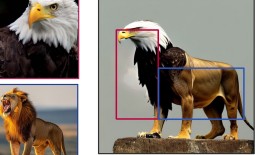

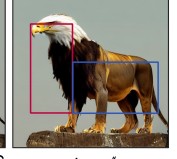

**Figure 6: Dynamic layout update** – While coarse boxes allow specific placements of content within the image, constraining attention-sharing to only retrieving content from the corresponding input image can lead to artifacts (e.g., brown patches outside the mask for dog and background artifacts around eagle). We resolve this by allowing the masks to be automatically adjusted.

dynamically update them during the generation process by a segmentation that finds the boundary of the generated subject.

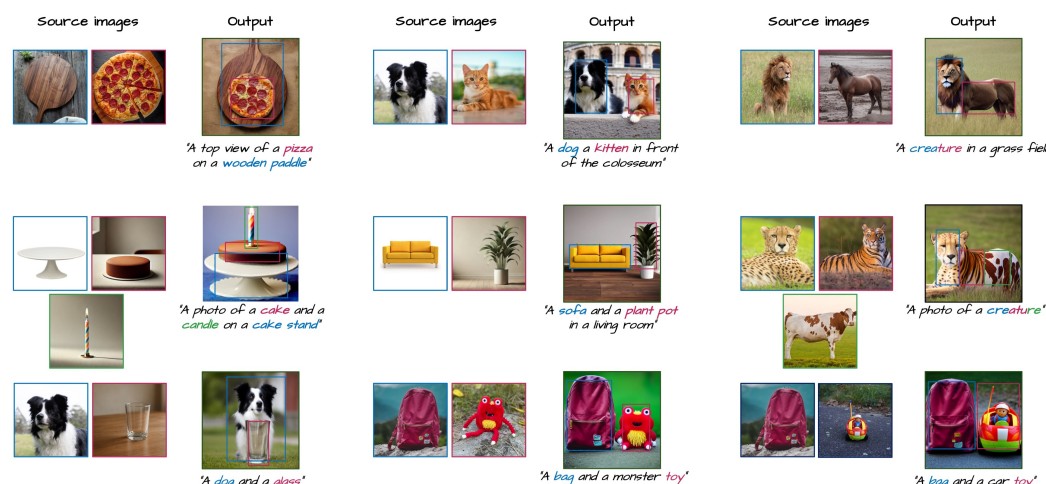

**Figure 7: Visual gallery** – We demonstrate our method's ability to perform various compositions.

For denoising at time step $t$, the noisy image $I_T^t$ is first denoised to produce the predicted clean image $\hat{I}_T^0$, which is then re-noised to obtain $I_T^{t-1}$. We realized that $\hat{I}_T^0$ is already a good approximation of the image and can be used by SAM (Kirillov et al., 2023) for object or part segmentation.

Since SAM requires prompt points for each region, we need a method to extract keypoints. We leverage features from the Diffusion's U-Net (DIFT) as they encode rich semantic information, useful for establishing point correspondences between images (Tang et al., 2023; Luo et al., 2023). Combining them with DINO (Caron et al., 2021) features further enhances correspondence (Zhang et al., 2023a). Therefore, we use this approach for layout-based keypoint detection, extracting DIFT features (Tang et al., 2023) from the U-Net during both source inversion and target image generation, and computing DINO features for the source and predicted clean target image $\hat{I}_T^0$. The final feature map used for keypoint detection is defined as:

$$F = \beta \cdot \text{norm}(F_{\text{DIFT}}) \oplus (1 - \beta) \cdot \text{norm}(F_{\text{DINO}}) \tag{9}$$

where $F$ denotes feature maps, $\text{norm}(\cdot)$ a normalization operation, and $\beta{=}0.5$ is a scaling parameter. To find the correspondence of a pixel $p$ in the target image $I_T$ with the source images $I_S$, we group pixels based on a similarity metric and select a representative set of pixels from each group as keypoints. Formally, we compute:

$$C_{T \to S}(p) = \arg \max_{q \in I_S} \cos \text{sim}(F_T(p), F_S(q)), \tag{10}$$

where $\cos \text{sim}(\cdot)$ represents cosine similarity, $F_T$ is the feature map of $I_T$, and $F_S$ denotes the feature maps of $I_S$. We then discard pixels in $I_T$ with low similarity scores using OTSU thresholding (Otsu, 1979) and group the remaining pixels into sets $\mathbf{G} = \{G_1, G_2, \ldots, G_N\}$ based on their highest-scoring correspondence in the source images. For each group $G_i$, we retain the top $R\%$ of pixels with the highest similarity scores and apply farthest-point sampling to extract $k$ keypoints, forming the set $K_i$. Finally, the set of per subject keypoints $\mathbf{K} = \{K_1, K_2, \ldots, K_N\}$ are fed into SAM to generate the updated layout masks. An overview of this process is depicted in Figure 5. details of our implementation can be found in the supplementary material.

## 5 EXPERIMENTS AND RESULTS

In this section, we first demonstrate the versatility of our approach by showcasing object-level and part-level composition results in various settings. Then, we conduct ablation studies to evaluate the contribution of each component of our approach and validate our design choices. We finally compare our method with several personalization and layout control approaches, demonstrating its effectiveness through quantitative metrics and a user study.

**Qualitative results.** Figure 7 presents a visual gallery of our results across different settings. Our method performs both object-level and part-level composition while respecting the layout arrangement, reference identities, and the input text prompt. The layout arrangement is flexible, allowing

**Table 1: Comparison of training time and user study results.** Prior methods require expensive fine-tuning, whereas Griffin is training-free and rated highest by users.

| Method | Training time (min) | User study ($\uparrow$) |
|---|---|---|
| TI Gal et al. (2022) + BA Dahary et al. (2024) | $\sim$25 | 1.54 |
| DB Ruiz et al. (2023) + BA Dahary et al. (2024) | $\sim$10 | 2.21 |
| Cones2 Liu et al. (2023) | $\sim$45 | 1.45 |
| MuDI Jang et al. (2024) + BA Dahary et al. (2024) | $\sim$100 | 1.99 |
| **Griffin (Ours)** | **0** | **3.22** |

for overlapping and non-overlapping boxes. We achieve this by assuming an order for the boxes and, as a preprocessing step, subtracting the front boxes from the back boxes.

**Ablations.** We present a visual ablation study in Figure 8. Omitting IP-Adapter initialization introduces artifacts. Removing attention sharing leads to identity and detail loss (e.g., hat pattern, cat's eye color), color leakage (e.g., cat's forehead), and reduced capability for part-level editing (e.g., teddy bear and Lego). Finally, dynamic masking prevents content and background leakage. We also provide a user study quantitatively verifying the effectiveness of our components in the supplementary.

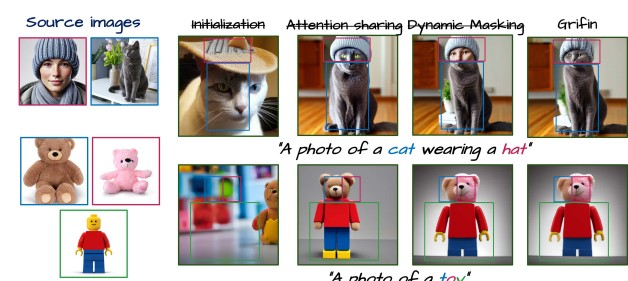

**Figure 8: Qualitative ablation –** Removing any component of our method results in artifacts showing the importance of all parts.

**Comparison.** As explained in Section 1, very few existing methods natively support both personalization and layout control. Therefore, we construct our comparison baselines by combining multiple personalization approaches with the state-of-the-art layout control method, Bounded attention (BA) (Dahary et al., 2024). We employ the following personalization methods: Textual Inversion (TI) (Gal et al., 2022), DreamBooth (DB) (Ruiz et al., 2023), and MuDI (Jang et al., 2024). MuDI supports multi-concept personalization by cutting and mixing subjects during training. For TI and DB, we fine-tune the text tokens or diffusion model weights for each reference image separately. We also include Cones2 (Liu et al., 2023), which supports both multi-concept personalization and layout control. Visual results of our comparison are shown in Figure 9. Overall, BA can mostly localize the subjects. But for identity preservation, TI often fails to maintain subject identity because the learned text token has limited representational power, while DB suffers from appearance leakage across different subjects. MuDI and Cones2 cannot reliably learn multiple subjects when there is only a single image per subject. Finally, none of the baselines can handle part-level composition effectively. Furthermore, in Table 1, we compare the training time of the methods.

In the supplementary, we also discuss MultiWine (Tarrés et al., 2025), which is a recent training-based approach that requires a curated dataset. Since neither data, code, nor model weights are available, we were not able to perform a thorough comparison. Instead, we provide visual comparisons using available images in the paper. Compared to MultiWine, Griffin is training-free, it better preserves identity, and additionally supports part-level composition.

**User study.** We conducted a user study to validate the quality of our results against competing methods, (see Table 1). Participants were presented with reference images, a layout, a text prompt, and outputs from Griffin and four alternatives. They ranked each output based on (1) layout accuracy, (2) identity preservation, and (3) text-prompt alignment. Responses from 30 participants across 25 examples indicate a strong preference for our method.

**Quantitative results.** We also run a quantitative comparison between our method and comparing baselines. we crop each layout component in the target image and we compare each crop with

their corresponding source images using DINOv2 (Oquab et al., 2024) and DreamSim (Fu et al., 2023) similarity metrics. For object-level composition, we use OWLv2 (Minderer et al., 2024) to extract each subject's bounding box. However, we found that for part-level examples OWL cannot extract correct part bounding boxes. Therefore, we use the input layout to crop the images. Since all the methods perform reasonably well for localization, we find this approach fair. The results are presented in Table 2.

**Table 2:** Quantitative comparison. Our method outperforms other baselines on similarity metrics.

| Method | DreamSim (↑) | DINOv2 (↑) |
|---|---|---|
| TI Gal et al. (2022) + BA Dahary et al. (2024) | 0.44 | 0.50 |
| DB Ruiz et al. (2023) + BA Dahary et al. (2024) | 0.52 | 0.59 |
| Cones2 Liu et al. (2023) | 0.44 | 0.50 |
| MuDI Jang et al. (2024) + BA Dahary et al. (2024) | 0.52 | 0.60 |
| Griffin | **0.57** | **0.61** |

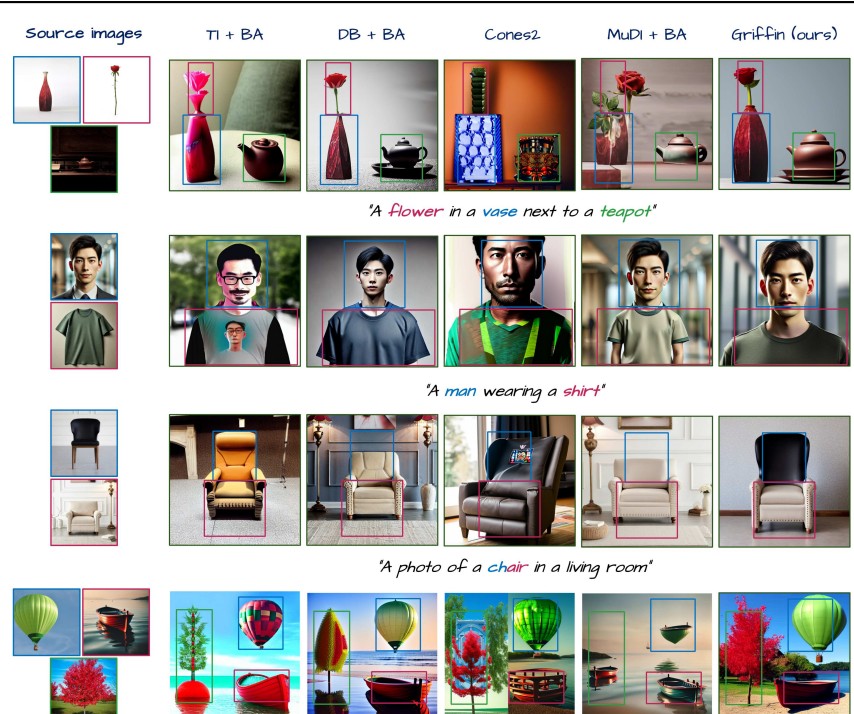

**Figure 9: Visual comparison** –Our method better captures the subjects' identity and composes them without artifacts or leakage.

## 6    CONCLUSION, LIMITATIONS, AND FUTURE WORK

We introduce Griffin, which offers a method for image composition by enabling part-level control and layout specification. By combining attention-sharing with layout control, it successfully maintains the identity of subjects while allowing for flexible placement within the generated scene. Griffin offers an efficient way to integrate elements from different images. With only one reference image per subject, Griffin outperforms existing techniques, providing a robust tool for both object-level and part-level composition tasks. The results demonstrate its effectiveness in producing realistic, and cohesive images. Through qualitative and quantitative experiments, user studies, and ablation studies, we showed the effectiveness of our method and its components. While our approach supports flexible composition with single-image references and requires no fine-tuning, it also has limitations. Since our attention-sharing mechanism copies the exact style from source images, it cannot perform text-based stylization or combine images with different styles. Adapting attention-sharing to support style transfer is a promising research direction. Such compositions over 3D objects' textures and geometry can also be an interesting avenue to explore for future work.

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

## A  EXTRA RESULTS

We provide extra visual results in Figure 10.

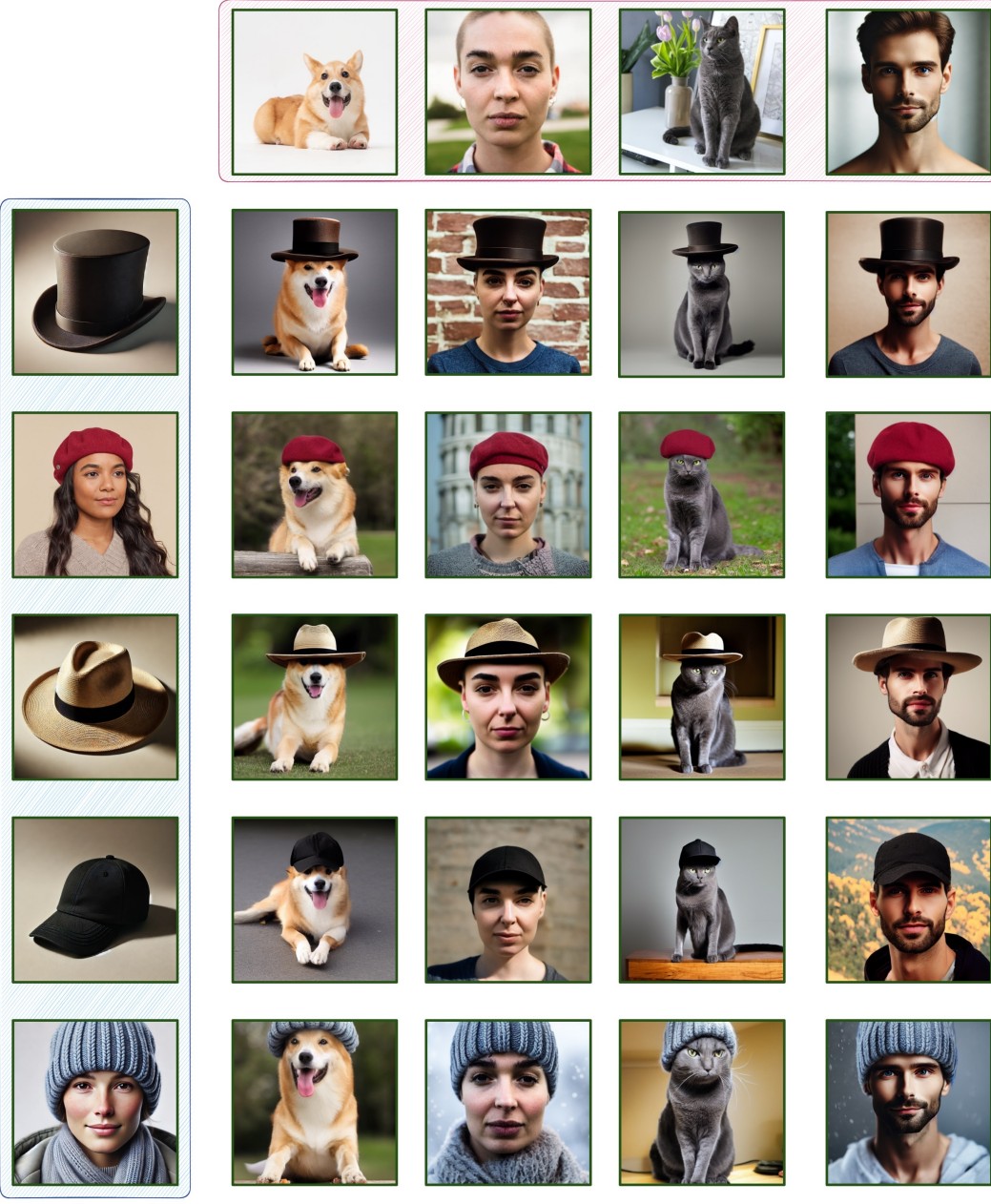

**Figure 10: More visual results.**

## B  IMPLEMENTATION DETAILS

For both inversion and generation, we use the DDIM scheduler with 50 timesteps. In the generation process, the first 10 steps are dedicated to structure initialization, while attention-sharing is applied during $t \in [10, 50]$. The IP-Adapter scale is set to 1.8 in the initialization stage. Then it is reduced to 0.8 and is later decreased to 0.4 in timestep 30. The value of $\alpha$ in Equation (6) is calculated using a scheduler with the function:

$$\alpha = \frac{1.2}{1 + 2e^{-10t}}, \tag{11}$$

assuming that in the denoising process $t \in [t_{LBA}, 0]$. This means that at the early timesteps of attention sharing the target attends more to the source images. As the generation progresses, the target attends more to itself. We empirically found that using this scheduler helps mitigate background artifacts. In our keypoint detection algorithm, we typically set $R$ to 50% and $k$ to 5. The dynamic mask update is performed at timesteps $t \in \{15, 20, 25, 30\}$.

While our method operates in a zero-shot manner without requiring fine-tuning of the diffusion model or textual inversion, we found that a short fine-tuning of the IP-Adapter's cross-attention key and value projection weights improves identity preservation (see inset). When applied, fine-tuning runs for 400-1000 steps and takes 3-6 minutes on a single RTX 3090 GPU.

## C FINETUNING IP-ADAPTER

To more effectively preserve the fine-grained details of each object, we optionally fine-tune the IP-Adapter for each subject in our experiments. We run this fine-tuning process separately for every object, using a masked variant of IP-Adapter to focus the loss on the specific region of interest. In particular, we obtain a binary mask for each subject and apply the loss only on its corresponding pixels (i.e., the region we wish to personalize). We employ the AdamW Loshchilov & Hutter (2019) algorithm for optimization, using a learning rate of 1e-4 and a weight decay of 1e-2. We also adopt a Direct Consistency Optimization (DCO) Lee et al. (2024) loss term to help the model remain close to its pretrained weights. In the original DCO framework, the loss terms are defined as follows:

$$\ell(\theta) = \|\epsilon_\theta(z_t; c, t) - \epsilon\|_2^2, \quad \ell(\phi) = \|\epsilon_\phi(z_t; c, t) - \epsilon\|_2^2$$

where $\epsilon_\theta$ is the fine-tuned model and $\epsilon_\phi$ is the reference model without LoRA. In our application, we instead disable the IP-Adapter for $\ell(\phi)$, ensuring that the baseline remains purely the unmodified pretrained model.

The DCO loss is then computed as:

$$L_{\text{DCO}}(\theta) = -\log \sigma \left(-\beta_t(\ell(\theta) - \ell(\phi))\right)$$

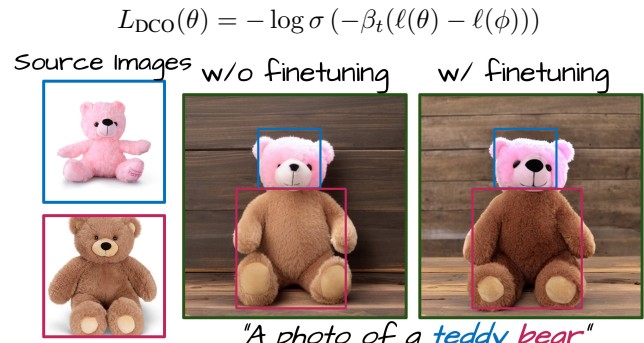

Figure 11: **Finetuning IP-Adapter** helps preserve finegrained details of the references.

By minimizing the KL divergence between these two losses, we constrain the network's drift away from the pretrained distribution. Empirically, this leads to slightly cleaner and more robust results.

## D COMPARISON WITH MULTIWINE

As noted in the main paper, MultiWine (Tarrés et al., 2025) is a recent work on multi-concept localized generation. Their method trains an image adapter and injects image features through cross-attention while fine-tuning a Stable Diffusion inpainting model. Since their approach relies on a curated dataset and neither code, models, nor data are publicly available, we instead provide visual comparisons using several examples from their paper for which the source images are publicly accessible. As shown in Figure 12, Griffin achieves higher identity preservation and produces more natural, realistic images while being completely training-free. We view MultiWine as a strong encoder-based personalization approach, and hypothesize that incorporating our localized attention-sharing and dynamic masking into its pipeline could further enhance identity fidelity.

**Figure 12: Comparison with MultiWine (Tarrés et al., 2025).** As evident in the results, the subjects' identities are better captured in those produced by Griffin. In Multiwine's outputs, additional artifacts, such as Einstein's beard, appearance, and object details are not well preserved, for instance the gold frames of the sunglasses, which are altered in both the Einstein and dog examples.

## E    ABLATIONS USER STUDY

To better assess the contribution of each component, we conducted a user study comparing our full method against two ablated variants: (i) using only masked IP-Adapter without attention-sharing, and (ii) disabling dynamic masking. We omit the case without IP-Adapter initialization, as its results were significantly worse and not informative. Following the same protocol as the user study in Section 5, participants were shown outputs from Griffin and from the corresponding ablated variant, and asked to select the better result based on (1) layout accuracy, (2) identity preservation, and (3) text-prompt alignment. Responses from 19 users across 10 examples are summarized in Table 3. The results confirm that Griffin is consistently preferred over masked IP-Adapter alone, and that dynamic masking substantially reduces content leakage and improves object placement.

**Table 3: Ablative user study.** Our full method is consistently preferred over variants without attention-sharing and dynamic masking.

| Ablation | Preference percentage (↑) |
|---|---|
| **Griffin** vs. Masked IP Adapter | 90.00% |
| **Griffin** vs. no Dynamic Masking | 64.21% |

## F    OTHER ARCHITECTURES

While our implementation is based on Stable Diffusion v1.5, the method can be extended to other architectures that (1) include an encoder-based personalization adapter (e.g., IP Adapter) and (2) incorporate self-attention blocks. In Figure 13, we present qualitative results on the SDXL (Podell et al., 2023) model, and in Figure 14, we demonstrate the extension of Griffin on FLUX-dev 1.0 (Labs, 2024), which is a DiT architecture.

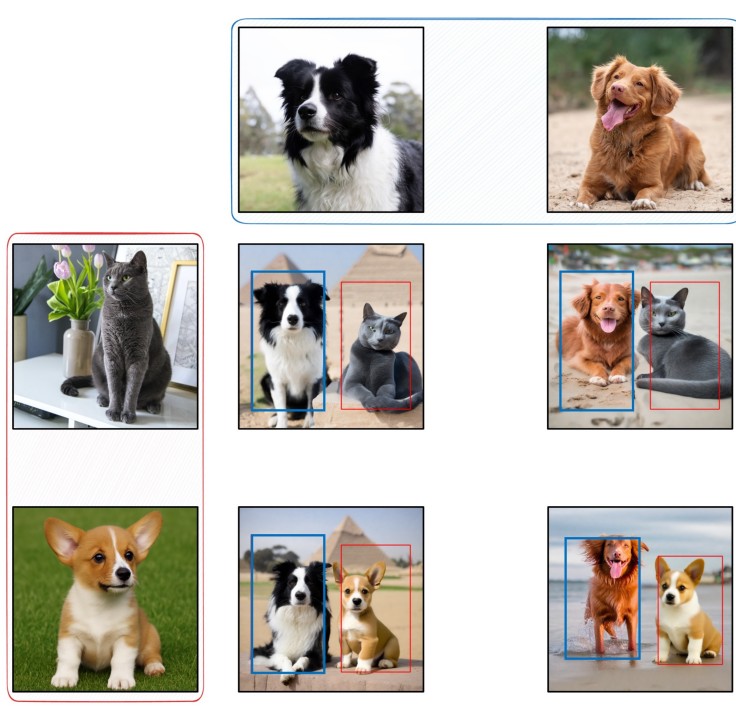

"… in the pyramids of Giza"          "… in a beach"

**Figure 13: Visual results on SDXL.** Our method is applicable to the SDXL diffusion architecture. By applying Griffin, we achieve personalized and localized image generation.

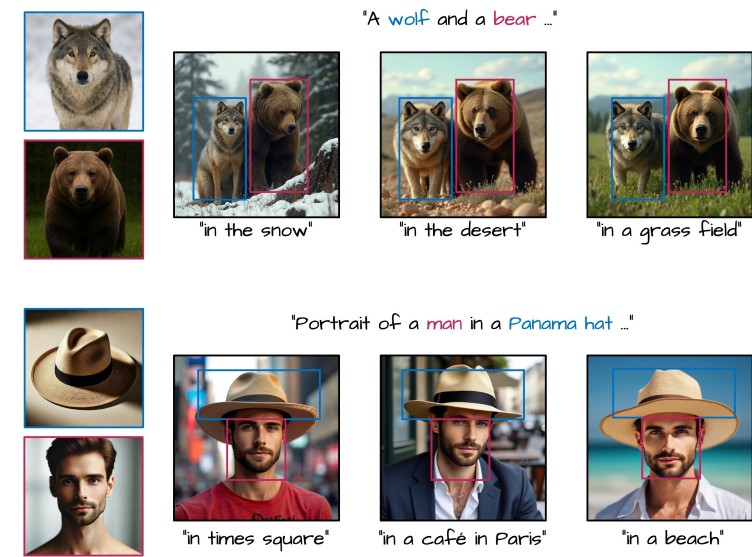

**Figure 14: Visual results on Flux.** Our method is applicable to the Flux DiT architecture. By applying Griffin, we achieve personalized and localized image generation.

# G    STATEMENT ON REPRODUCABILITY AND LLM USAGE

Code of our method is attached as a supplementary to the submission and will be publicly available upon acceptance. Please note that we used ChatGPT for minor rephrasing to avoid grammar issues.

