# OpenReview forum: "Griffin: Generative Reference and Layout Guided Image Composition"
_ICLR.cc/2026/Conference — ICLR 2026 Conference Withdrawn Submission_

### Official Review · Reviewer_tdFD · 2025-10-29

**Soundness:** 3
**Presentation:** 2
**Contribution:** 2
**Rating:** 6
**Confidence:** 4

**Summary:**

This paper addresses a clear and important problem in generative AI: precise spatial and identity control. The method combined existing techniques (IP-Adapter, attention-sharing) in a clever and effective way.  The results are impressive and the paper is generally well-written. However, all core modules mentioned in this paper are borrowed, which could be considered lack of novelty in general. It still has several weaknesses that must be addressed before it can be considered for acceptance.

**Strengths:**

1. Practical and Training-Free: Eliminates the need for per-concept fine-tuning or textual inversion, which is a significant usability and efficiency advantage.
2. The use of DIFT/DINO + SAM to update masks during generation is clever and effective.
3. This paper demonstrated on SD 1.5, SDXL, and FLUX (DiT), suggesting it has broad applicability.

**Weaknesses:**

1. The paper claims to be “training-free,” yet the best results use 3–6 minutes of per-subject IP-Adapter fine-tuning. This should be clarified: is Griffin truly zero-shot, or is fine-tuning essential for high fidelity?

2. The method assumes disjoint or ordered masks. How does it scale to dozens of objects? Runtime or memory overhead of caching multiple K/V per denoising step is not discussed.

3. Limited novelty in core components: Attention sharing (or MasaCtrl), IP-Adapter, DIFT, SAM, and bounded attention are all borrowed. The novelty lies in their integration and scheduling, which is solid but incremental.

4. As acknowledged, Griffin copies style exactly and cannot stylize references per prompt (e.g., “a Van Gogh-style dog”). This limits creative control.

5. While extensions to SDXL and FLUX are shown in the appendix (Figures 13-14), the main results are based on SD v1.5, which is now relatively outdated.

**Questions:**

1.The paper shows successful cases. Can you provide a dedicated section or figure discussing common failure modes? For example, when does the dynamic mask update fail? How does the method handle significant differences in lighting/perspective between source images?

2.Can the method handle more than 3-4 reference images simultaneously?

3.What is the typical inference time compared to standard SD generation?

Suggestions

1. Clarify whether IP-Adapter fine-tuning is optional or recommended for best results.

2. Include a runtime/memory analysis vs. baselines.

3. Show failure cases (e.g., ambiguous layouts, style conflicts).

4. Provide a pseudo-code algorithm box for the full pipeline

---

### Official Review · Reviewer_C6Mi · 2025-10-31

**Soundness:** 3
**Presentation:** 3
**Contribution:** 2
**Rating:** 6
**Confidence:** 4

**Summary:**

GRIFFIN is a training-free image composition method that combines IP-Adapter for structural guidance with "layout-controlled attention sharing" to achieve high-fidelity content combination. This approach uses only single reference images and layout information to precisely control the placement and appearance of objects or parts in a new image, while using dynamic masking to prevent identity leakage and edge artifacts. Extentive experiments show the performance of Griffin.

**Strengths:**

- The method is simple and easy to reproduce, which  requires no training.
- The paper is clearly written and well-structured.
- Extensive experiments are conducted to demonstrete the overall performance.

**Weaknesses:**

1. The training-free claim is misleading as it ignores prohibitive inference costs. The method requires DDIM inversion plus multiple calls to large models (DIFT, DINO, SAM), making its single-run cost potentially higher than competitors' fine-tuning.
2. The Dynamic Layout Update pipeline (DIFT/DINO/SAM) is a fragile serial pipeline. The paper fails to analyze its failure modes, where an error in any step could cascade.
3. Experiments are limited to simple 2-3 object compositions, raising scalability concerns. The paper fails to address potential bottlenecks as N increases (N>5) or performance in complex scenes with occlusion.
4. The comparisons are insufficient, using personalization methods (TI, DB) as baselines. The paper must compare against relevant  methods like InstanceDiffusion (Wang et al., 2024) and MIGC (Zhou et al., 2024).Even if it's just for visual presentation.

- Wang, X., Darrell, T., Rambhatla, S. S., Girdhar, R., & Misra, I. (2024). InstanceDiffusion: Instance-level Control for Image Generation.
- Zhou, D., Li, Y., Ma, F., Zhang, X., & Yang, Y. (2024). MIGC: Multi-instance generation controller for text-to-image synthesis.

**Questions:**

Coud you provide results of objects larger objects (N>5) and the performance in complex scenes with occlusion?
Even if you have put some results with Flux in Appendix, why not take it as main results?

---

### Official Review · Reviewer_jrPi · 2025-11-01

**Soundness:** 2
**Presentation:** 3
**Contribution:** 2
**Rating:** 4
**Confidence:** 5

**Summary:**

This paper presents Griffin, a training-free method for generative reference and layout guided image composition. The approach enables precise control over both content and placement in image generation by conditioning on reference images rather than relying solely on text prompts. Griffin consists of three main stages: (1) structure initialization using masked IP-Adapter to align layout regions with reference images, (2) layout-controlled attention sharing to preserve identity while preventing cross-reference leakage, and (3) dynamic layout update using SAM segmentation to refine masks during generation. The method supports both object-level and part-level composition with only a single reference image per subject, and demonstrates superior performance compared to existing approaches through extensive experiments, user studies, and ablation analyses.

**Strengths:**

- The training-free approach and single-reference requirement represent significant practical advantages over fine-tuning-based methods, eliminating computational burdens while maintaining reasonable performance.
- The integration of multiple existing techniques into a cohesive framework demonstrates thoughtful engineering and system design.
- The ability to extend the method to different diffusion architectures (SDXL, FLUX-dev) demonstrates a degree of architectural robustness.

**Weaknesses:**

- This work offers limited innovation, as the advances are primarily incremental. Its principal contribution appears to lie in the integration of pre-existing methods.
- The current approach primarily focuses on preserving the identity of reference images rather than enabling creative style transfer or stylistic variation.
- The three-stage process, particularly the dynamic mask update with feature extraction and SAM segmentation, adds significant computational overhead.

**Questions:**

1. The core technical components lack sufficient innovation, with cross-image attention for reference injection previously explored in FreeCustom[1] and attention-based layout control investigated in DenseDiffusion[2] and related works. The authors need to clarify the differences between the proposed method and these works.
2. The authors do not mention how the value of the hyperparameter β is chosen, nor do they analyze the impact of the proportion of DIFT features and DINO features on the effectiveness of dynamic masking. Is it necessary to include both features?
3. What is the computational complexity of the dynamic mask update mechanism, and how does it scale with the number of reference images and their resolution?
4. Can the proposed method handle dense interactions between subjects, such as the natural overlap and mutual penetration of body parts when a couple embraces?
5. How robust is the feature correspondence method to variations in object pose, viewpoint, and scale between reference and target images?

[1] Ding G, Zhao C, Wang W, et al. Freecustom: Tuning-free customized image generation for multi-concept composition[C]//Proceedings of the IEEE/CVF Conference on Computer Vision and Pattern Recognition. 2024: 9089-9098.

[2] Kim Y, Lee J, Kim J H, et al. Dense text-to-image generation with attention modulation[C]//Proceedings of the IEEE/CVF International Conference on Computer Vision. 2023: 7701-7711.

---

### Official Review · Reviewer_gL7G · 2025-11-02

**Soundness:** 2
**Presentation:** 3
**Contribution:** 2
**Rating:** 2
**Confidence:** 3

**Summary:**

The paper presents Griffin, a personalized image generation that allows multi-image layout control, where one can define content through images and the placement of the content in the final generated image. This is done in a training-free method, which involves structure initialization, attention sharing and a dynamic layout update step. Through qualitative examples and quantitative results, which include a user study and similarity metrics such as DinoV2, the authors present the effectiveness of the method in comparison to existing methods.

**Strengths:**

1. Compared to existing personalized methods, Griffin is training-free which makes it efficient.
2. The paper is well-written and easy to follow.

**Weaknesses:**

1. Missing clarity on results and baselines :

i) What was the evaluation set used in Table 1? What is the distribution of "object-level" and "part-level" personalization? Further, details about annotation agreement etc. are missing.

ii) Existing benchmarks such as https://arxiv.org/pdf/2401.13388 exist, on which no results have been presented.

iii) In addition to Multitwine, there exist other methods such as Emu2 (https://arxiv.org/pdf/2312.13286), VisualComposer (https://arxiv.org/pdf/2501.01424) and PiT (https://arxiv.org/pdf/2503.10365, especially for part-based generation) which are baselines that should be considered.

iv) Most personalized methods also report CLIP-I and CLIP-T scores, these numbers should also be reported to understand full generalization of the proposed approach.

2. I am not fully convinced on the motivation of the work. I do believe that bounding boxes / layouts are not the best form of inputs in this case. Does the model even need the masks in these cases and is text not a better form of input in this case? For example , Figure 6, instead of a "A photo of a creature", a prompt such as. "A creature with the head of an eagle and the body of a lion" -- describes the image better. This will also largely simplify the method as components such as dynamic layout update would not be needed in that case.

3. Overall, the paper feels to be overtly complex, without compelling results. Case in point being, the method has a bunch of hyperparamaters (alpha in Eq. 6, s in structure initialization, beta in equation 9), uses a lot of pre-existing methods such as SAM, DINO, DIFT, IP-Adapter etc. This also makes one question the generalization of the proposed approach and the overall effect on inference speed.

**Questions:**

Please refer to weaknesses.

---

### Note · Authors · 2025-11-13

I have read and agree with the venue's withdrawal policy on behalf of myself and my co-authors.